# Resource Allocation in Two-Patch Epidemic Model with State-Dependent Dispersal Behaviors Using Optimal Control

**Sunmi Lee** [1] **, Okbun Baek** [1] **and Luis Melara** [2],*****

[1]  Department of Applied Mathematics, Kyung Hee University, Yongin 446-701, Korea;
    sunmilee@khu.ac.kr (S.L.); obbaek1@gmail.com (O.B.)
[2]  Department of Mathematics, Shippensburg University, 1871 Old Main Drive, Shippensburg, PA 17257, USA
*****  Correspondence: lamelara@ship.edu

**Abstract:** A two-patch epidemic model is considered in order to assess the impact of virtual dispersal on disease transmission dynamics. The two-patch system models the movement of individuals between the two-patches using a residence-time matrix $P$, where $P$ depends on both residence times and state variables (infected classes). In this work, we employ this approach to a general two-patch SIR model in order to investigate the effect of state dependent dispersal behaviors on the disease dynamics. Furthermore, optimal control theory is employed to identify and evaluate patch-specific control measures aimed at reducing disease prevalence at a minimal cost. Optimal policies are computed under various dispersal scenarios (depending on the different residence-time matrix configurations). Our results suggest there is a reduction of the outbreak and the proportion of time spent by individuals in a patch exhibits less fluctuations in the presence of patch-specific optimal controls. Furthermore, the optimal strategies for each patch differ depending on the type of dispersal behavior and the different infection rate in a patch. In all of our results, we obtain that the optimal strategies reduce the number of infections per patch.

**Keywords:** two-patch model with virtual dispersal; the basic reproduction number; final epidemic size; optimal control interventions

## 1. Introduction

Modeling the transmission of diseases has been studied over the past decades in a variety of forms, see [1–14]. The mathematical formulation of these cases is determined by a combination of the real-world situation at hand and mathematical tools deemed suitable. Some previous work, such as [15–17], serve as a guide in this paper. The mathematical modeling of the spread of diseases is important as the movement of people and other living organisms increases due to globalization. Real-world examples of public health concerns included the potential threat of the spread of the Zika virus following the Rio 2016 Olympics in Brazil, when thousands of humans worldwide traveled to Brazil and then returned to their home countries [18]. A commonly posed question was: whether the visitors became infected with Zika during their time in Brazil, would their return home spread Zika outside of Brazil? A worldwide Zika spread did not happen, of course. On the other hand, in January 2020, the coronavirus, COVID-19, with origins in Wuhan, China did spread to 214 countries. COVID-19 has resulted in 1,689,724,318 confirmed cases and 663,470 fatalities as of 30 July 2020 and its impact ranged from countries' public health systems to their economies. However, this scenario raises a general question about the potential worldwide spread of diseases, as humans and other living organisms are no longer restricted by man-made and natural boundaries. To address this question,

mathematical models that involve multiple patches have been developed representing disconnected geographical locations and the consequences of infectious organisms traveling between these patches.

We use a simple case involving only two populations each residing in physically separate locations, for instance, rural versus urban. For simplicity, we assume travel for both populations is restricted between their two home locations only. Supposing a disease outbreak occurs, we now pose the following questions:

- How does the disease spread in the different locations with intermixing populations?
- How does the disease outbreak affect the behavior of residents and visitors in both locations?
- How should preventive resources be optimally allocated in different locations to reduce the number infections?

The mathematical study and analysis of multi-patch models can be found in [3,15,16]. The authors investigated the impact of travel between patches for spatial spread of disease in the Eulerian framework (a mobility-matrix approach) [3]. They obtained the relation between the global $R_0$ and the local $R_0$ in their multi-patch model with different level of disease prevalence. In [15], the authors present and analyze multi-patch models with their basic reproductive numbers in the Lagrangian framework (a residence-matrix approach). In [16], optimal strategies for two-patch dengue transmission is numerically studied via optimal control problems. As this is a theoretical model where we approximate solutions computationally, we refer to the travel of populations between two locations, as virtual dispersal in the Lagrangian framework based on [15]. In our report, we present an optimal control formulation for two-patch SIR models under virtual dispersal where the control function represents interventions or policy for personal protection (such as face masks, disinfectants, sanitizers, and etc.). As we are not considering space in our mathematical model, we interpret the amount of time spent in either location as a representation of time physically spent this location (similar to contact tracing using GPS in smart phones).

This paper is organized, as follows. In Section 2, we present a two-patch SIR model with virtual dispersal, discuss the basic reproduction number and formulate the associated optimal control problem. Section 3 will show numerical results from approximating solutions to the optimal control problem. The paper concludes with a discussion of results in Section 4. Following Section 4, we include an Appendix A containing the mathematical work and proofs showing the existence of the adjoint variables and the characterization of the optimal control functions.

## 2. A Two-Patch Sir Model with Virtual Dispersal

The motivation for the two-patch model is rooted in the SIR model without demographic dynamics proposed by Kermack and McKendrick in 1927 [4], and is given by

$$\begin{aligned}
\dot{S}(t) &= -\beta S(t)\,I(t) \\
\dot{I}(t) &= \beta S(t)\,I(t) - \alpha I(t) \\
\dot{R}(t) &= \alpha I(t).
\end{aligned} \tag{1}$$

where $S(t)$ represents the susceptible, $I(t)$ the infected, and $R(t)$ the recovered populations at time $t$ with constant population $N = S(t) + I(t) + R(t)$.

An underlying assumption in (1) is that the epidemiological dynamics are restricted to one physical location. This location is called a patch. The two-patch model considers the scenario where two populations reside in two physically disjoint locations; each patch with its own set of epidemiological dynamics also governed by their own version of (1), but restricted to their own patch. The two-patch model further assumes that members from each patch will spend portions of their time in only two locations: their own patch or the other patch. This information is incorporated into the mathematical model using a residence-time matrix. This interaction introduces its own set of epidemiological dynamics: being infected by or infecting members of their own patch or the other patch and how much

time members will spend in their own patch or the other patch. The two-patch model in this report was previously studied for SIS and SIR models in [15]. Figure 1 presents a compartmental model that depicts the two-patch model and Section 2.1 discusses the mathematical model in more detail.

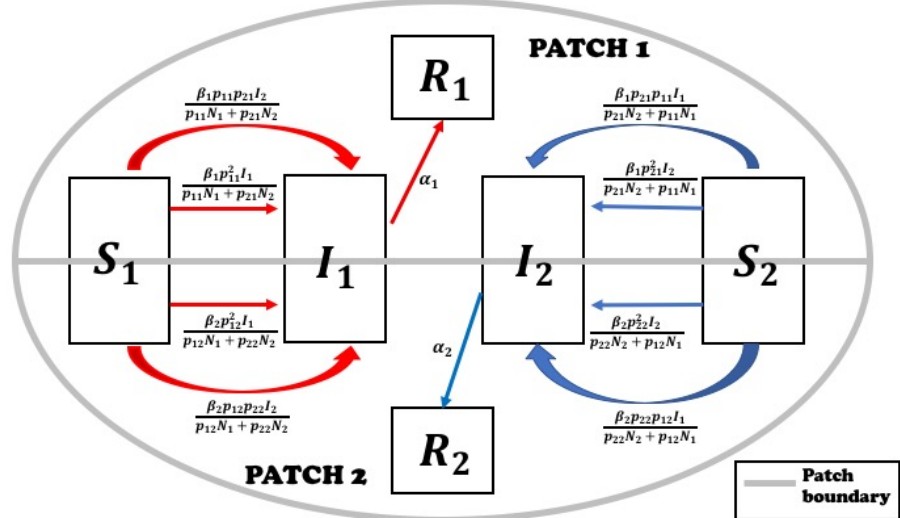

**Figure 1.** Basic two-patch SIR compartmental model with virtual dispersal and no demographic dynamics. This compartmental model is mathematically described in (3). For visual simplicity, the patches are juxtaposed. We note that patches 1 and 2 may not share a border.

### 2.1. A Two-Patch Model with Virtual Dispersal

The two-patch SIR mathematical model applies (1) to a patch $i$ with $i = 1, 2$. Each patch $i$ consists of three epidemiological classes; $S_i(t)$ for the susceptible class, $I_i(t)$ for the infected class and $R_i(t)$ for the recovered class with the constant population $N_i = S_i(t) + I_i(t) + R_i(t)$. For convenience, we write $S_i = S_i(t)$, $I_i = I_i(t)$ and $R_i = R_i(t)$. Both patches are coupled via a residence-time matrix, $P = (p_{ij}) \in \mathbb{R}^{2 \times 2}$, where the $p_{ij} = p_{ij}(I_1, I_2)$ represent the proportion of time that a person residing in patch $i$ spends in patch $j$ with $\sum_{j=1}^{2} p_{ij} = 1$, $i = 1, 2$. Note that this residence-time matrix, $p_{ij}(I_1, I_2)$ is a function of $I_1(t)$ and $I_2(t)$. This residence-time matrix captures behavioral responses by modeling that the proportion of time spent in a particular patch depends on the numbers of infected individuals on that particular patch; that is $P = P(I_1, I_2)$.

The mathematical construction of $p_{ij}(I_1, I_2)$, summarized in [15], satisfies the following conditions:

$$\frac{\partial p_{11}(I_1, I_2)}{\partial I_1} \le 0 \quad \text{and} \quad \frac{\partial p_{22}(I_1, I_2)}{\partial I_2} \le 0. \tag{2}$$

This suggests that as the population of $I_i$ increases, then the proportion of time spent by members $S_i$ and $I_i$ in their respective patch $i$ decreases. In short, the conditions (2) mathematically describe a behavior response to an increase in the infected population $I_i$ in patch $i$. Below, we present the functions $p_{ij}$ that satisfy (2):

$$p_{11}(I_1, I_2) = \frac{\sigma_{11} + \sigma_{11} I_1 + I_2}{1 + I_1 + I_2}, \qquad p_{12}(I_1, I_2) = \sigma_{12}\left(\frac{1 + I_1}{1 + I_1 + I_2}\right),$$

$$p_{21}(I_1, I_2) = \sigma_{21}\left(\frac{1 + I_2}{1 + I_1 + I_2}\right), \quad p_{22}(I_1, I_2) = \frac{\sigma_{22} + I_1 + \sigma_{22} I_2}{1 + I_1 + I_2}$$

with $\sum_{j=1}^{2} \sigma_{ij} = 1$, $i = 1, 2$ and $\sigma_{ij}$ values are summarized in Table 1.

**Table 1.** Baseline parameter values $\sigma_{ij}$ in residence-time matrix $P$, $i, j = 1, 2$.

| Dispersal Scenarios | Residence-Time Proportions |
|---|---|
| Polar | $\sigma_{11} = \sigma_{22} = 1$, $\sigma_{12} = \sigma_{21} = 0$ |
| Symmetric | $\sigma_{ij} = 0.5$, $\forall i, j$ |
| Asymmetric | $\sigma_{11} = 0.6$, $\sigma_{22} = 0.9$, $\sigma_{12} = 0.4$, $\sigma_{21} = 0.1$ |
| High-mobility | $\sigma_{11} = \sigma_{22} = 0$, $\sigma_{12} = \sigma_{21} = 1$ |
| Uni-directional 1 | $\sigma_{11} = 1, \sigma_{22} = 0$, $\sigma_{12} = 0, \sigma_{21} = 1$ |
| Uni-directional 2 | $\sigma_{11} = 0, \sigma_{22} = 1$, $\sigma_{12} = 1, \sigma_{21} = 0$ |

A susceptible individual from patch $i$ can be infected by infected individuals from patch $i$ or $j$ in proportion to the total number of individuals from both patches while all are residing in either patch $i$ or $j$. Hence, the incidence rate at which individuals from patch $i$ get infected by an infected individual also from patch $i$ is described next (over and under braces included for emphasis):

- For $S_1$ : $\left( \beta_1 p_{11} \times \overbrace{\dfrac{p_{11} I_1}{\underbrace{p_{11} N_1 + p_{21} N_2}_{\text{while in patch 1}}}}^{\text{proportion of } I_1} + \beta_2 p_{12} \times \overbrace{\dfrac{p_{12} I_1}{\underbrace{p_{12} N_1 + p_{22} N_2}_{\text{while in patch 2}}}}^{\text{proportion of } I_1} \right) \times S_1,$ and

- For $S_2$ : $\left( \beta_1 p_{21} \times \dfrac{p_{21} I_2}{p_{11} N_1 + p_{21} N_2} + \beta_2 p_{22} \times \dfrac{p_{22} I_2}{p_{12} N_1 + p_{22} N_2} \right) \times S_2.$

A susceptible individual from patch $i$ may also be infected by a proportion of individuals from patch $j$, while both are in either patch $i$ or $j$. This yields (again, using over and under braces for emphasis):

- For $S_1$ : $\left( \beta_1 p_{11} \times \overbrace{\dfrac{p_{21} I_2}{\underbrace{p_{11} N_1 + p_{21} N_2}_{\text{while in patch 1}}}}^{\text{proportion of } I_2} + \beta_2 p_{12} \times \overbrace{\dfrac{p_{22} I_2}{\underbrace{p_{12} N_1 + p_{22} N_2}_{\text{while in patch 2}}}}^{\text{proportion of } I_2} \right) \times S_1,$ and

- For $S_2$ : $\left( \beta_1 p_{22} \times \dfrac{p_{12} I_1}{p_{11} N_1 + p_{21} N_2} + \beta_2 p_{21} \times \dfrac{p_{11} I_1}{p_{12} N_1 + p_{22} N_2} \right) \times S_2.$

The two-patch dynamics are captured by the following ordinary differential equations

$$
\begin{aligned}
\dot{S}_i &= -\left[ \frac{\beta_i p_{ii}^2}{p_{ii} N_i + p_{ji} N_j} + \frac{\beta_j p_{ij}^2}{p_{ij} N_i + p_{jj} N_j} \right] S_i I_i - \left[ \frac{\beta_i p_{ii} p_{ji}}{p_{ii} N_i + p_{ji} N_j} + \frac{\beta_j p_{ij} p_{jj}}{p_{ij} N_i + p_{jj} N_j} \right] S_i I_j \\[2mm]
\dot{I}_i &= \left[ \frac{\beta_i p_{ii}^2}{p_{ii} N_i + p_{ji} N_j} + \frac{\beta_j p_{ij}^2}{p_{ij} N_i + p_{jj} N_j} \right] S_i I_i + \left[ \frac{\beta_i p_{ii} p_{ji}}{p_{ii} N_i + p_{ji} N_j} + \frac{\beta_j p_{ij} p_{jj}}{p_{ij} N_i + p_{jj} N_j} \right] S_i I_j - \alpha_i I_i \\[2mm]
\dot{R}_i &= \alpha_i I_i
\end{aligned}
\tag{3}
$$

where $i = 1, 2$. It suffices to solve for $S_i$ and $I_i$ using only the first two equations of (3) and then solve for $R_i$. For simplicity, in Section 2.3, the reduced state equations only involving the first two equations of (5) will be referred to as the state equations.

We explore the effects of virtual dispersal scenarios on the basic reproduction number $\mathcal{R}_0$ and the final epidemic size. Different virtual dispersal scenarios certainly change $\mathcal{R}_0$ and the final epidemic size, but determining whether the six different assumptions can change them substantially or not, is nontrivial. The level of transmissibility measured by $\mathcal{R}_0$ is varied to highlight the differences and similarities for the results under several virtual dispersal scenarios. We consider the following distinct virtual dispersal scenarios, as given in Table 1.

### 2.2. Basic Reproduction Number and Final Epidemic Size

One of the most important factors in mathematical epidemiology is the basic reproduction number. The basic reproductive number, $\mathcal{R}_0$, is the average number of secondary infectious cases when one

infectious individual is introduced in a wholly susceptible population. The next generation method is used to compute the basic reproduction number, $\mathcal{R}_0$ [19]. The basic reproduction number $\mathcal{R}_0$, mathematically, is the largest eigenvalue of the next generation matrix $K \in \mathbb{R}^{2\times 2}$ obtained below,

$$
K = -\mathcal{F}\mathcal{V}^{-1} = \begin{bmatrix} \left(\frac{\beta_1 p_{11}^2}{p_{11}N_1 + p_{21}N_2} + \frac{\beta_2 p_{12}^2}{p_{12}N_1 + p_{22}N_2}\right)\frac{N_1}{\alpha_1} & \left(\frac{\beta_1 p_{11}p_{21}}{p_{11}N_1 + p_{21}N_2} + \frac{\beta_2 p_{12}p_{22}}{p_{12}N_1 + p_{22}N_2}\right)\frac{N_1}{\alpha_2} \\ \left(\frac{\beta_1 p_{11}p_{21}}{p_{11}N_1 + p_{21}N_2} + \frac{\beta_2 p_{12}p_{22}}{p_{12}N_1 + p_{22}N_2}\right)\frac{N_2}{\alpha_1} & \left(\frac{\beta_1 p_{21}^2}{p_{11}N_1 + p_{21}N_2} + \frac{\beta_2 p_{22}^2}{p_{12}N_1 + p_{22}N_2}\right)\frac{N_2}{\alpha_2} \end{bmatrix}.
$$

Upon evaluation at the disease-free equilibrium, we obtain the global basic reproduction number

$$
\mathcal{R}_0 = \max\left\{ \text{eigenvalue}(K) \right\} \tag{4}
$$

In [15], the authors reported that not everybody is infected during an outbreak, and so estimating the size of the total infected population (the final epidemic size in the absence of deaths or departures) is tied in the solutions of the final size relationship. The residence time matrix $P$ plays an important role, as evidenced by the dependence of the final epidemic size relation. We compute the cumulative number of new infected cases (or the final epidemic size) numerically by solving the equation

$$
\dot{C}_i = \left[ \frac{\beta_i p_{ii}^2}{p_{ii}N_i + p_{ji}N_j} + \frac{\beta_j p_{ij}^2}{p_{ij}N_i + p_{jj}N_j} \right] S_i I_i + \left[ \frac{\beta_i p_{ii}p_{ji}}{p_{ii}N_i + p_{ji}N_j} + \frac{\beta_j p_{ij}p_{jj}}{p_{ij}N_i + p_{jj}N_j} \right] S_i I_j.
$$

The quantity $C_i(t_f)$ is used to compute the patch-specific final epidemic size for $i = 1, 2$.

### 2.3. Optimal Control Formulation

The optimal control problem of interest is formulated through the incorporation of the control functions $(1 - u_i)$ in the transmission rates for patch $i$ ($i = 1, 2$). The effect of these interventions implicitly reduces the transmission rates $\beta_i$, $i = 1, 2$. The two-patch model in this report was previously studied for SIS and SIR models in [15]. We extend the two-patch SIR model by incorporating control functions $0 \leq u_i(t) \leq 1$, where, for simplicity, we write $u_i = u_i(t)$, for $i = 1, 2$. The two-patch dynamics with patch-specific controls are captured by the following ordinary differential equations

$$
\dot{S}_i = -\left[ \frac{\beta_i(1-u_i)p_{ii}^2}{p_{ii}N_i + p_{ji}N_j} + \frac{\beta_j(1-u_j)p_{ij}^2}{p_{ij}N_i + p_{jj}N_j} \right] S_i I_i - \left[ \frac{\beta_i(1-u_i)p_{ii}p_{ji}}{p_{ii}N_i + p_{ji}N_j} + \frac{\beta_j(1-u_j)p_{ij}p_{jj}}{p_{ij}N_i + p_{jj}N_j} \right] S_i I_j
$$

$$
\dot{I}_i = \left[ \frac{\beta_i(1-u_i)p_{ii}^2}{p_{ii}N_i + p_{ji}N_j} + \frac{\beta_j(1-u_j)p_{ij}^2}{p_{ij}N_i + p_{jj}N_j} \right] S_i I_i + \left[ \frac{\beta_i(1-u_i)p_{ii}p_{ji}}{p_{ii}N_i + p_{ji}N_j} + \frac{\beta_j(1-u_j)p_{ij}p_{jj}}{p_{ij}N_i + p_{jj}N_j} \right] S_i I_j - \alpha_i I_i \tag{5}
$$

$$
\dot{R}_i = \alpha_i I_i
$$

Via an optimal control formulation, the terms $(1 - u_i)$ in (5) can effect the transmission rates in the $i$th patch. Accordingly, if $u_i = 0$, then there is no control and the transmission rate remains the same. As $u_i \to 1$, then we approach an "ideal efforts" situation thereby reducing the transmission rate to 0. Again, the preventive control efforts may involve face masks, disinfectants, social distancing, education campaigns, and so forth with the intention of increasing personal protection. Our goal is to minimize the infected individuals in both patches at a minimal cost of implementation over a finite time horizon via optimal control. The objective functional to be minimized is

$$
J(u_1, u_2) = \int_0^{t_f} I_1 + I_2 + \frac{1}{2}\left( W_1 u_1^2 + W_2 u_2^2 \right) dt
$$

and subject to the state equations in (5). The constants $W_1$ and $W_2$ are the weights for the prevention effort or the relative cost of the implementation of the preventive control for patch 1 and patch 2,

respectively. The objective function will both minimize the number of infective people and the levels of cost of prevention. We seek an optimal pair $(U^*, X^*)$, such that

$$J(U^*) = \min_{U \in \Omega} J(U), \tag{6}$$

with $U = (u_1, u_2)$, and where $\Omega = \{U \in (L^1(0, t_f))^2 | \, 0 \leq u_i(t) \leq 1, t \in [0, t_f], i = 1, 2\}$ subject to the state system (5) with $X = (S_1, I_1, R_1, S_2, I_2, R_2)$. The existence of optimal controls is guaranteed from standard results on optimal control theory [20]. Pontryagin's Maximum Principle is used to establish necessary conditions that must be satisfied by an optimal solution [21]. Derivations of the necessary conditions are shown in the Appendix A. Additionally, we compute the cumulative number of new infected cases in the presence of controls by solving the equation

$$\dot{C}_i = \left[ \frac{\beta_i(1 - u_i)p_{ii}^2}{p_{ii}N_i + p_{ji}N_j} + \frac{\beta_j(1 - u_j)p_{ij}^2}{p_{ij}N_i + p_{jj}N_j} \right] S_i I_i + \left[ \frac{\beta_i(1 - u_i)p_{ii}p_{ji}}{p_{ii}N_i + p_{ji}N_j} + \frac{\beta_j(1 - u_j)p_{ij}p_{jj}}{p_{ij}N_i + p_{jj}N_j} \right] S_i I_j.$$

The quantity $C_i(t_f)$ is used to compute the patch-specific controlled final epidemic size for $i = 1, 2$.

## 3. Numerical Results

Numerical solutions to (A1) were obtained using the standard scheme (a two point boundary method [22]), which is employed, as follows. First, the state system (5) is solved forward in time with initial conditions and an initial guess for the control. Second, the adjoint system (A9) with transversality conditions (see Theorem A1 in Appendix A) is solved backward in time. Third, the optimality condition is updated using the characterization formula (A10), see Theorem A1 in Appendix A. These three steps are iterated until convergence is achieved. Parameter values are given in Tables 1 and 2.

**Table 2.** Baseline parameter values for each patch.

| Parameter | Description | Value |
|---|---|---|
| $\beta_1$ | Transmission rate in patch 1 (days$^{-1}$) | 0.3–0.4 |
| $\beta_2$ | Transmission rate in patch 2 (days$^{-1}$) | 0.5–0.6 |
| $\alpha_1$ | Recovery rate in patch 1 (days$^{-1}$) | 0.25 |
| $\alpha_2$ | Recovery rate in patch 2 (days$^{-1}$) | 0.25 |
| $N_1$ | Population size in patch 1 | 1000 |
| $N_2$ | Population size in patch 2 | 1000 |
| $S_1(0)$ | The initial value of susceptible in patch 1 | 999 |
| $S_2(0)$ | The initial value of susceptible in patch 2 | 999 |
| $I_1(0)$ | The initial value of infected in patch 1 | 1 |
| $I_2(0)$ | The initial value of infected in patch 2 | 1 |
| $t_f$ | The simulated duration (days) | 60 |
| $b$ | The upper bound of control | 0.5 |
| $W_1$ | Weight constant corresponding to control $u_1$ | 100–300 |
| $W_2$ | Weight constant corresponding to control $u_2$ | 100–300 |

Table 1 shows the $\sigma_{ij}$ values when the model does not incorporate state dependence [15]. In particular, we note that $\sigma_{ij} = p_{ij}(0, 0), \, \forall i, j$ [15]. The numerical values for the polar, symmetric, asymmetric, and high-mobility dispersal scenarios were obtained from [15]. In this paper, we also introduce the Uni-directional 1 and Uni-directional 2 scenarios, which were determined by inspection from the surface plot of $\mathcal{R}_0$ as a function of $(\sigma_{11}, \sigma_{22})$. The Uni-directional 1 dispersal scenario represents all of the members of patch 1 remaining in and all members of patch 2 traveling to patch 1 during an epidemic outbreak. Similarly, the Uni-directional 2 dispersal scenario represents all members of patch 1 traveling to and all members of patch 2 remaining in patch 2 during an epidemic outbreak.

We assume the population sizes $N_1 = N_2$ and that $\beta_1 < \beta_2$ where $\beta_1$ and $\beta_2$ values can be found in Table 2. The mobility patterns that are described by the residence-time matrix $(p_{ij})$ suggest different

virtual dispersal between the two patches. We use the $\sigma_{ij}$ values in Table 1 to describe the different types of coupling between patches. The polar case suggests that the populations remain put in their home patches. In the symmetric scenario, the proportion of humans visiting from patch *i* to patch *j* is the same as the other way around. Asymmetric mobility implies that a larger proportion of humans remain in their home patch than the other patch, with the second patch residents staying at home the longest. Finally, in the high mobility scenario, a higher proportion of humans visit the other patch as opposed to staying in their home patch. In the following Sections 3.1–3.3, we present the results in the absence of controls and the presence of controls, respectively.

### 3.1. Results in the Absence of Controls

In this section, we compare the population plots of solutions obtained from solving the two-patch SIR system (3) in the absence of controls. Figures 2–4 show the prevalence plots for the Polar/High Mobility, Symmetric/Asymmetric, and Uni-directional1/2 cases, respectively.

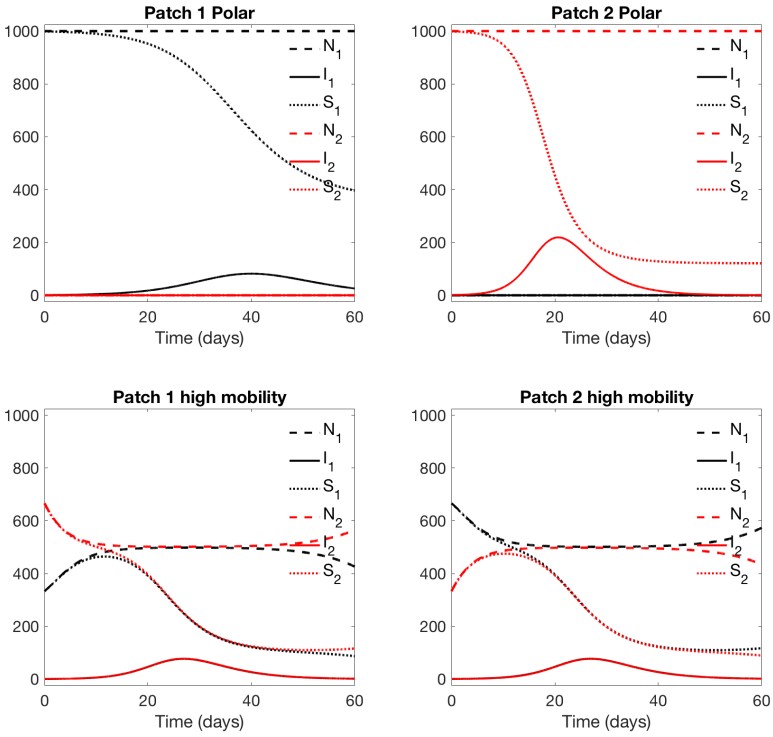

**Figure 2. Top row:** Polar case. **Bottom row:** High Mobility case. **Left column:**. Comparison of Patch 1 and 2 prevalence while members from both patches reside in Patch 1. **Right column:** Comparison of Patch 1 and 2 prevalence comparisons while from both patches reside in Patch 2. In *Patch 1 polar plot*, the graphs of $S_2$, $I_2$ and $N_2$ overlap. In *Patch 2 polar plot*, the graphs of $S_1$, $I_1$ and $N_1$ overlap. In *Patch 1 and 2 high mobility* plots , the graphs of $I_1$ and $I_2$ overlap.

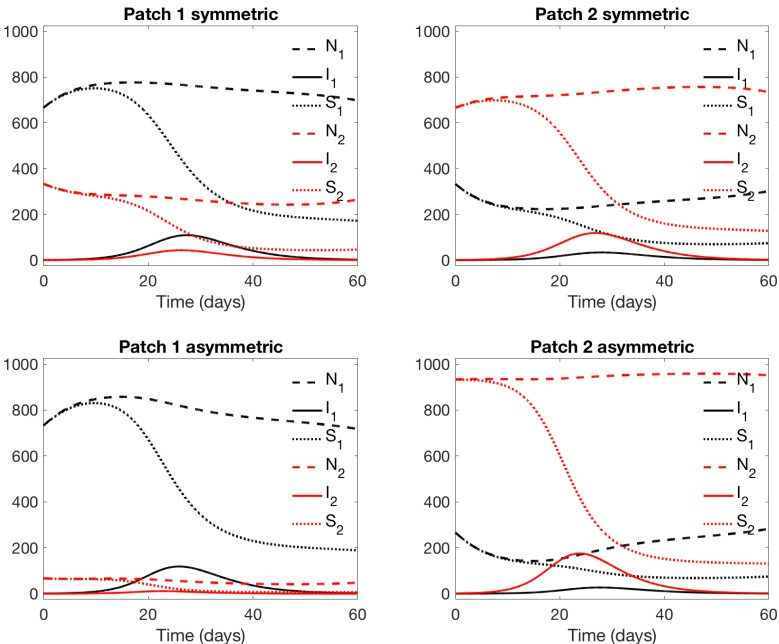

**Figure 3. Top row:** Symmetric case. **Bottom row:** Asymmetric case. **Left column:** Comparison of Patch 1 and 2 prevalence while members from both patches reside in Patch 1. **Right column:** Comparison of Patch 1 and 2 prevalence while members from both patches reside in Patch 2.

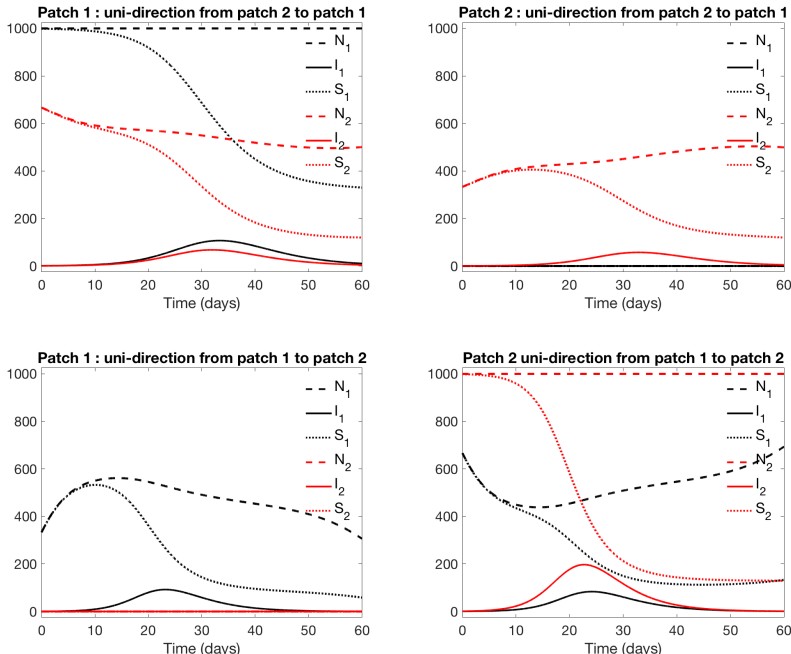

**Figure 4. Top row:** Unidirectional case from patch 2 to patch 1. **Bottom row:** Unidirectional case from patch 1 to patch 2. **Left column:.** Comparison of Patch 1 and 2 prevalence while members from both patches reside in Patch 1. **Right column:** Comparison of Patch 1 and 2 prevalence while members from both patches reside in Patch 2. In *Patch 2: uni-direction from patch 2 to patch 1* plot, the graphs of $S_1$, $I_1$ and $N_1$ overlap. In *Patch 1: uni-direction from patch 1 to patch 2* plot, the graphs of $S_2$, $I_2$ and $N_2$ overlap.

As with Figures 2–4, each row corresponds to a coupling case and all of the graphs show the plots of the proportions of members from patch 1 and 2 on the patch they are currently situated. For instance,

the curve labeled $p_{11}I_1$ represents the proportion of infected individuals from patch 1 who remain in patch 1, whereas the curve labeled $p_{21}I_2$ represents the proportion of infected individuals from patch 2 currently visiting patch 1 and so forth. The populations plotted are the susceptible $S_i$, infected $I_i$, and total $N_i$, $i = 1, 2$. The set of plots on the left half describe the populations in patch 1 and the set of plots on the right half describe the populations in patch 2.

In Figure 2, the polar case suggests that the spread of the disease was only confined to home residents who remained within their home patch; there was no travel between patches and therefore, no spread of disease from visitors. The total population remained constant throughout the duration of the epidemic. In the high mobility case, the number of infected individuals from either patch is the same for both patches, in spite of the number of visiting susceptibles being higher. The visiting susceptible population drops as low as the resident susceptible population. Additionally, the plots suggest that the overall number of visitors drops in either patch and possibly return to their home patch at the onset of epidemic, remain in their home patch and only later on return again to visit the other patch. In Figure 3, the symmetric case shows what may be interpreted as a small exodus of home residents and visitors from patch 1 to patch 2. In the asymmetric case, the patch 2 residents stay in patch 2, while the patch 1 individuals travel to patch 2. As we can notice in Figure 4, the prevalence in low risk patch 1 is highest in the uni-directional case 2 where as in high risk patch 2, uni-directional case 1 leads to the lowest prevalence.

Next, we present the impact of dispersal scenarios on the global $\mathcal{R}_0$ and final epidemic size under three different sets of $\beta_1$ and $\beta_2$. Figure 5 shows the surface that corresponds to the $\mathcal{R}_0$ values as a function of $\sigma_{11}$ and $\sigma_{22}$. We note that the surface is greater than 1 for all values of $\sigma_{11}$ and $\sigma_{22}$. The plot also labels the location for $\mathcal{R}_0$ values corresponding to the polar, symmetric, asymmetric and high mobility cases. As $\mathcal{R}_0$ increases, the spread of the epidemic also increases. Based on the locations of the four coupling cases along the surface, the disease will spread the most in the polar case when there is no prevention in place.

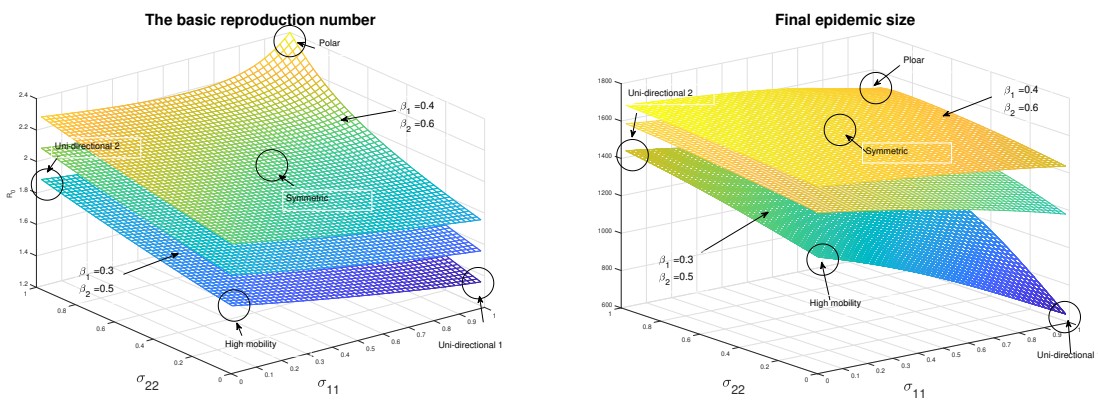

**Figure 5.** The global basic reproduction number, $\mathcal{R}_0$ and the final epidemic size is plotted as a function of $\sigma_{11}$ and $\sigma_{22}$ under three different sets of $\beta_1$ and $\beta_2$. Note that polar, symmetric, asymmetric, high-mobility, and uni-directional scenarios are labeled (maximum at polar and minimum at uni-directional case).

Overall values of $\mathcal{R}_0$ and the final epidemic size increase as $\beta_1$ and $\beta_2$ increase (three layers), as shown in Figure 5. Note that polar, high-mobility, symmetric, and uni-directional scenarios are labeled. The left panel of Figure 5 displays the global $\mathcal{R}_0$ given in (4) as functions of $\sigma_{11}$ and $\sigma_{22}$. Recall that the population sizes $N_1 = N_2$ and $\beta_1 < \beta_2$; $\mathcal{R}_0$ gets the minimum at the unidirectional case 1 ($\sigma_{11} = 1$ and $\sigma_{22} = 0$) while $\mathcal{R}_0$ gets the maximum at the polar case when $\sigma_{11} = 1$ and $\sigma_{22} = 1$.

The right panel of Figure 5 illustrates the impact of dispersal scenarios on the final epidemic size, which is also displayed as a function of $\sigma_{11}$ and $\sigma_{22}$. It is worth noting that the final epidemic size gets the minimum at the unidirectional case 1 ($\sigma_{11} = 1$ and $\sigma_{22} = 0$) while it has the largest at the

unidirectional case 2 ($\sigma_{11} = 0$ and $\sigma_{22} = 1$). For the unidirectional case 1, since dispersal is only from Patch 2 (higher risk) to Patch 1 (lower risk), the total final epidemic size in Patch 1 and Patch 2 gets the minimum among different dispersal scenarios. On the other hand, for the unidirectional case 2, since dispersal is only from Patch 1 (lower risk) to Patch 2 (higher risk), the total final epidemic size gets the maximum. The minimum is consistent with the results of $\mathcal{R}_0$; however, different results for the maximum (the polar case was the largest). This implies that $\mathcal{R}_0$ is not always the same as the final epidemic size. Lastly, Figure 6 highlights the final epidemic size as a function of $\beta_1$ and $\beta_2$ under four different dispersal scenarios. For all cases, obviously, the final epidemic size increases as $\beta_1$ and $\beta_2$ increase.

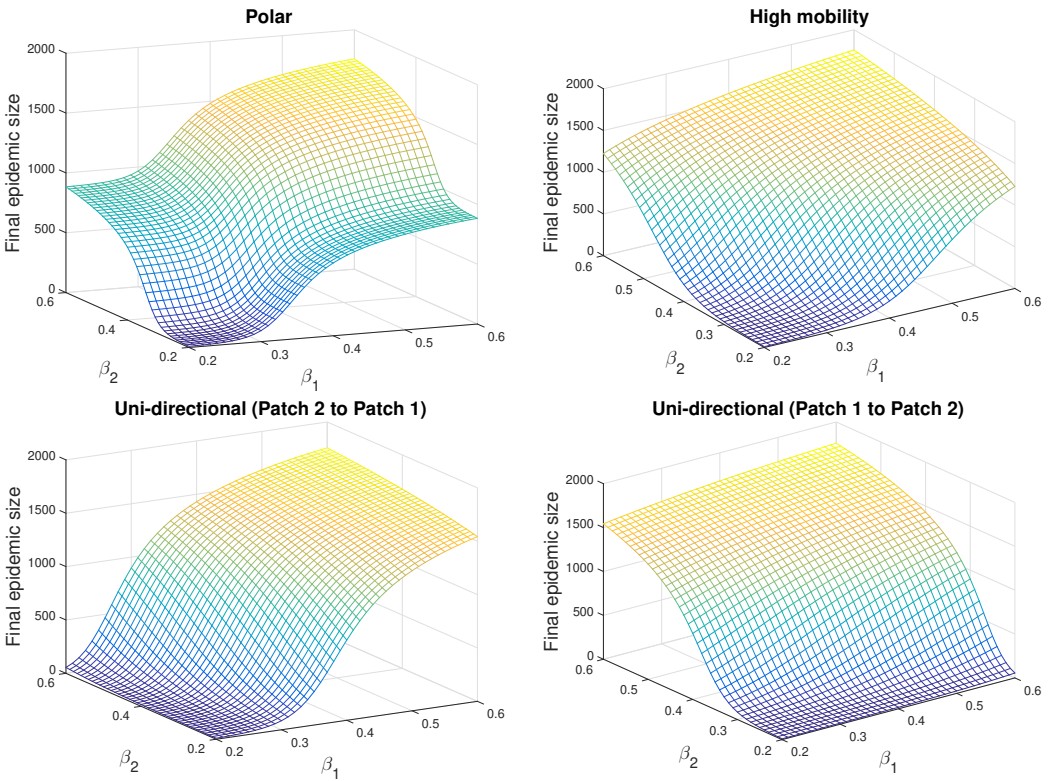

**Figure 6.** The final epidemic size is displayed as a function of $\beta_1$ and $\beta_2$ under four different cases. Overall final epidemic size increases as $\beta_1$ and $\beta_2$ increase.

### 3.2. Results in the Presence of Controls

In this section, we compare the population plots of solutions obtained from solving optimal control problem (A1) to solutions without any control functions. Figures 7–9 show the control and prevalence plots for the Polar/High Mobility, Symmetric/Asymmetric, and Uni-directional1/2 cases, respectively.

For all dispersal scenarios, the control plots suggest that the preventive measures start at the maximum level early on and then be reduced. Additionally, these plots suggest that the preventive measures remain at the maximum level the longest for patch 2 (recall $\beta_1 < \beta_2$). This becomes more pronounced in polar and uni-directional case 2 (see Figures 7 and 9). In almost all instances, the preventive measures remain at the highest level for the duration of the time interval. However, the polar case is differs the most. In the polar case, the preventive measures in patch 1 remain at the highest level for less than half the time in patch 2. Additionally, the preventive measures for patch 2 are in place the longest for the polar case. This may be attributed to both populations not traveling outside their home patch. Therefore, when populations are not traveling upon the onset of the spread of a disease, the bulk of preventive resources should be devoted to patch 2. In the high mobility case, the preventive measures are in place at the highest levels for about the same amount of time as the symmetric and asymmetric cases. This suggests that any travel between patches will need for the

preventive measures to remain at their highest levels for about the same amount of time, regardless of the type of movement between patches.

The prevalence plots presented in Figures 7–9 compare the population sizes when preventive measures are in place or not. When prevention is present, the number of infected individuals drops significantly in all cases. This suggests that prevention does play a positive role in reducing the spread of infection. In the polar case, the number of patch 1 infected is the lowest of all cases and the number of patch 2 infected is the highest. This appears to suggest that since the populations don't travel between patches, the spread of the disease remains confined to each patch when recalling that $\beta_1 < \beta_2$. Travel between patches, on the other hand, appears to spread the disease more evenly among the populations from each patch.

We measured the total value of the objective functional and the cumulative incidence as functions of $\sigma_{11}$ and $\sigma_{22}$ in order to further see the impact of virtual dispersal. Figures 10 and 11 present the total value of the objective functional and the final epidemic size in the presence of controls. It is clear how optimal control strategies reduce the cumulative incidence; compare these with Figure 5 which corresponds to results without control. Figures 10 and 11 illustrate the results under two weight constants (low cost using $W_1 = W_2 = 100$ and high cost using $W_1 = W_2 = 300$). Figure 10 shows the results under the transmission rates $\beta_1 = 0.3$ and $\beta_2 = 0.5$. Figure 11 shows the results under the transmission rates $\beta_1 = 0.4$ and $\beta_2 = 0.6$. If the relative cost of control is higher, then the total value of the objective function and the cumulative incidence are proportionally higher as well (see the bottom panels) in both Figures 10 and 11. This implies that, as we increase the relative cost size, we obtain less controls and, therefore, larger final epidemic sizes. Regardless of the costs of control, having a large force of transmission makes an outbreak extremely expensive to control, as shown in Figure 11.

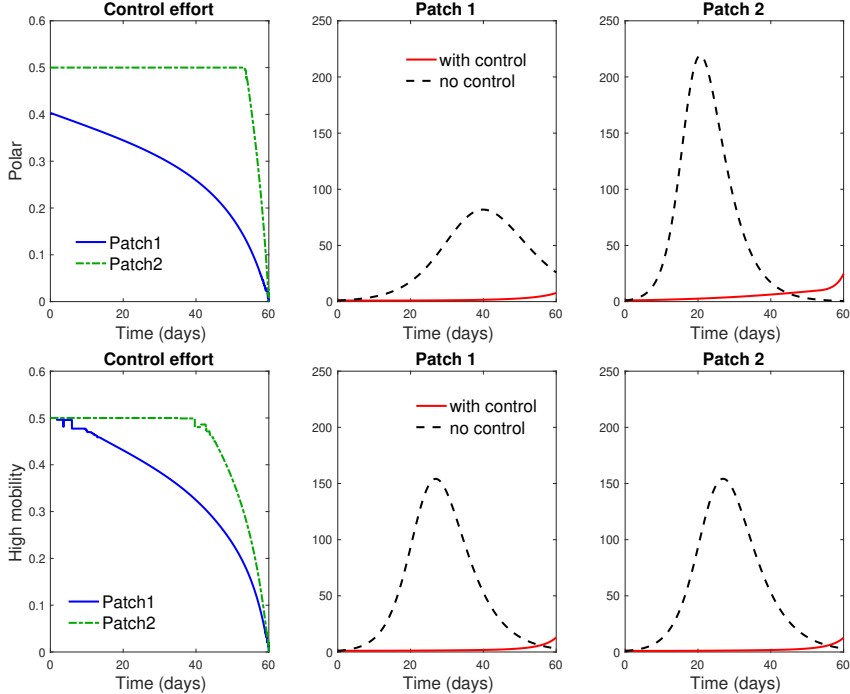

**Figure 7.** Top row: Polar case. Objective functional value at optimal control is $= J(u^*, v^*) = 1401.86$ and $\mathcal{R}_0 = 2.4$. **Bottom row: High Mobility case. Objective functional value at optimal control is $J(u^*, v^*) = 1317.53$ and $\mathcal{R}_0 = 2.0099$.** Patch-specific control functions, prevalence in Patch 1, and prevalence in Patch 2 are displayed in the left, middle, right panel, respectively. For each dispersal scenario, the corresponding Patch 1 and Patch 2 panels share the same legend.

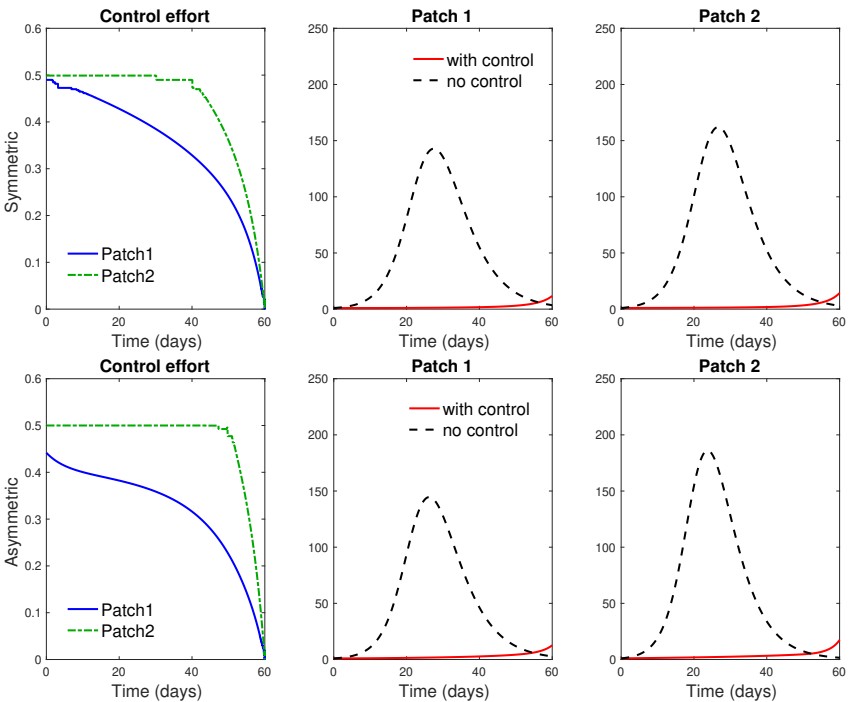

**Figure 8.** **Top row: Symmetric case. Objective functional value at optimal control is** $J(u^*, v^*) = 1303.25$ **and** $\mathcal{R}_0 = 2.0099$**. Bottom row: Asymmetric case. Objective functional value at optimal control is** $J(u^*, v^*) = 1385.12$ **and** $\mathcal{R}_0 = 2.137$**.** Patch-specific control functions, prevalence in Patch 1, and prevalence in Patch 2 are displayed in the left, middle, right panel, respectively. For each dispersal scenario, the corresponding Patch 1 and Patch 2 panels share the same legend.

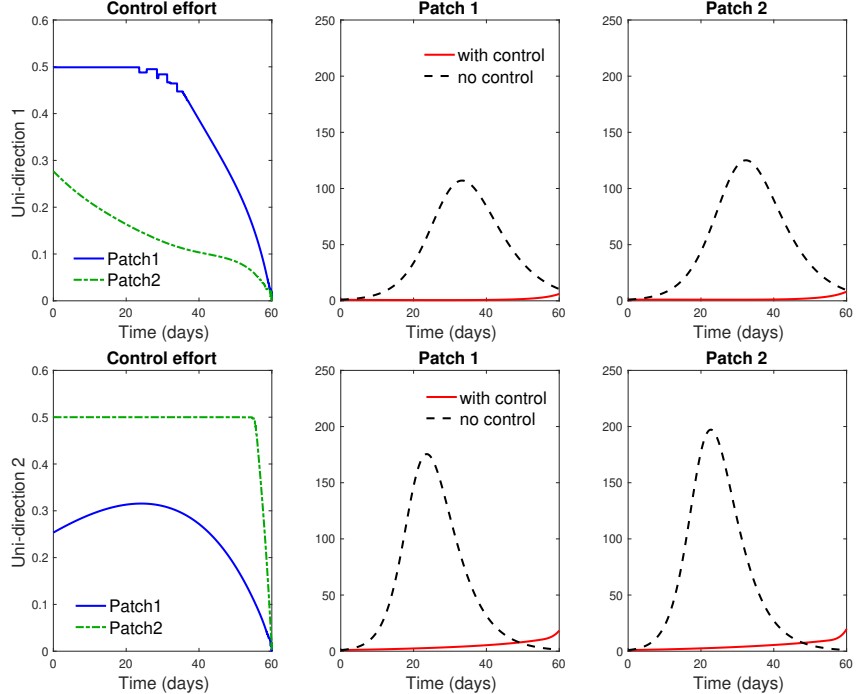

**Figure 9.** **Top row: Unidirectional case from patch 2 to patch 1. Objective functional value at optimal control is** $J(u^*, v^*) = 776.53$ **and** $\mathcal{R}_0 = 1.7471$**. Bottom row: Unidirectional case from patch 1 to patch 2. Objective functional value at optimal control is** $J(u^*, v^*) = 1492.79$ **and** $\mathcal{R}_0 = 2.2759$**.** Patch-specific control functions, prevalence in Patch 1, and prevalence in Patch 2 are displayed in the left, middle, right panel, respectively. For each dispersal scenario, the corresponding Patch 1 and Patch 2 panels share the same legend.



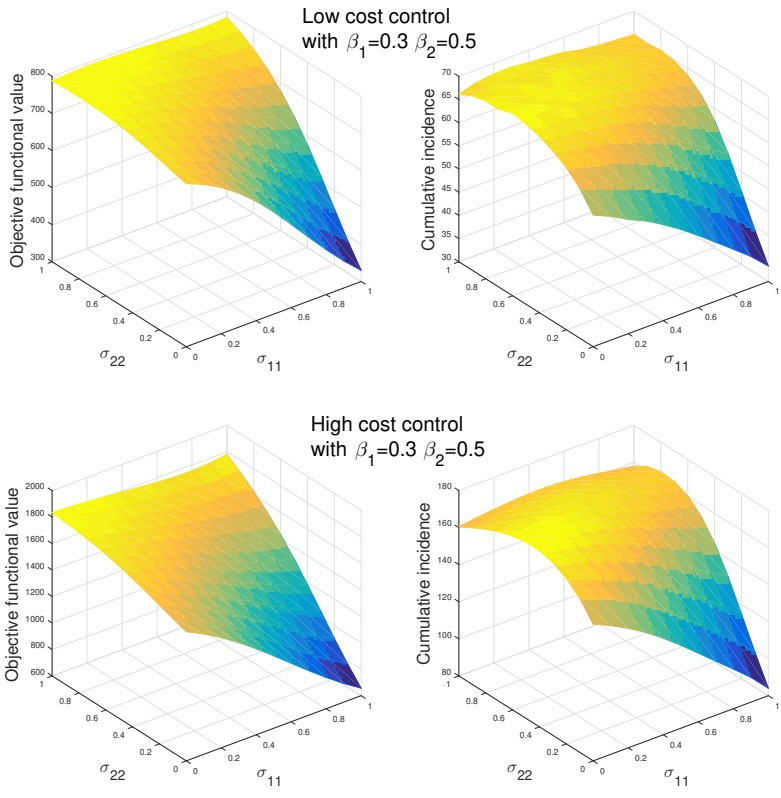

**Figure 10. Top row: low cost (**$W_1 = W_2 = 100$**). Bottom row: high cost (**$W_1 = W_2 = 400$**).** Objective functional value (left) and final epidemic size are displayed (right) using lower transmission rates ($\beta_1 = 0.3$ and $\beta_2 = 0.5$).

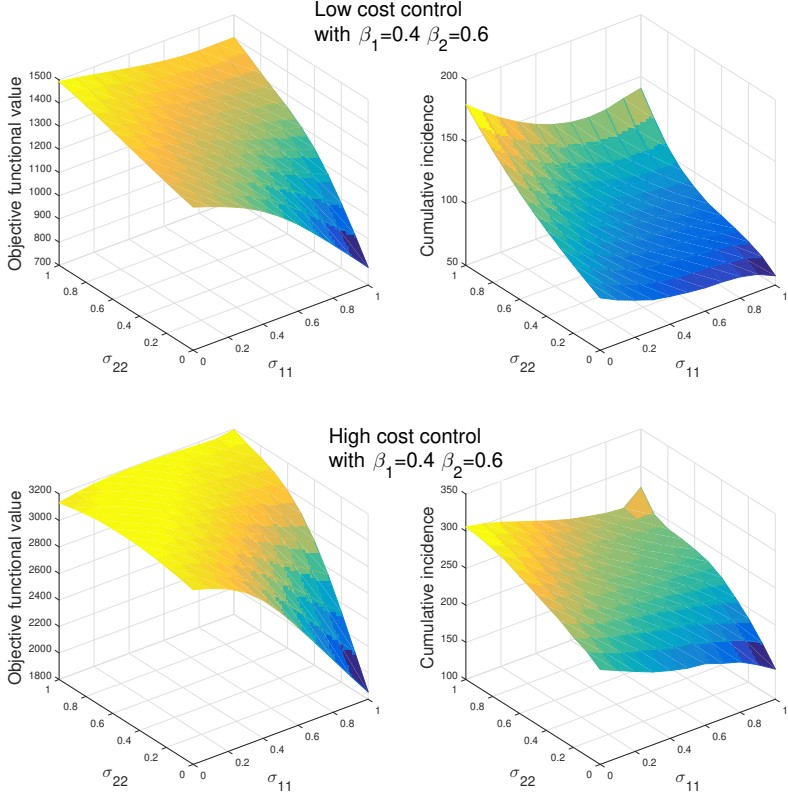

**Figure 11. Top row: low cost (**$W_1 = W_2 = 100$**). Bottom row: high cost (**$W_1 = W_2 = 400$**).** Objective functional value (left) and final epidemic size are displayed (right) using higher transmission rates ($\beta_1 = 0.4$ and $\beta_2 = 0.6$).

### 3.3. Various Control Scenarios

We use the objective functional (6) as a metric to compare the solution to the optimal control problem (A1) subject to (5), to the following intervention strategies: No control, one optimal control and maximum control. The 'No control' strategy corresponds no intervention implemented at all in both patches. The objective functional at this scenario is therefore represented by $J(0,0)$. The 'One optimal control' case represents the case where intervention is implemented in only patch but not the other. This yields two possibilities: intervention in patch 1, but no intervention in patch 2 or vice-versa; these are represented by $J(\bar{u}^*, 0)$ or $J(0, \bar{v}^*)$, respectively. We note that in the optimal solution $(\bar{u}^*, 0)$, the function $\bar{u}^*$ may not be the same as $u^*$ from $(u^*, v^*)$. This also holds for $\bar{v}^*$ and $v^*$. In the 'Max. control' scenario, we consider the maximum possible intervention is in place in both patches for all $t$. Because $u, v \in \Omega$ and $0 \leq u, v \leq 0.5$ over $[0, t_f]$, we set $u = 0.5$ and $v = 0.5, \forall t$. In particular, our optimal control solutions $(u^*, v^*)$ minimize (6). In all instances, our numerical results yielded the following relationships:

- No control: $J(u^*, v^*) < J(0,0)$
- One optimal control (two possibilities): $J(u^*, v^*) < J(\bar{u}^*, 0)$ or $J(u^*, v^*) < J(0, \bar{v}^*)$
- Maximum control: $J(u^*, v^*) < J(0.5, 0.5)$

Table 3 summarizes the percentage in reduction between $J(u^*, v^*)$ to evaluation of $J(u, v)$ at the optimal solutions corresponding to the above intervention strategies.

**Table 3.** Table of reduction values of objective functional (6) at optimal controls $(u^*, v^*)$ in comparison to various prevention strategies for the four coupling cases from Table 1.

| | Objective Functional Value Reduction Percentage | | | |
| --- | --- | --- | --- | --- |
| | **No Control** | **One Optimal Control** | | **Max. Control** |
| Dispersal scenarios | $J(0,0)$ | $J(\bar{u}^*, 0)$ | $J(0, \bar{v}^*)$ | $J(0.5, 0.5)$ |
| Polar | 315.4% | 177.5% | 137.9% | 29.7% |
| Symmetric | 382.8% | 287.0% | 149.4% | 23.9% |
| Asymmetric | 360.5% | 299.7% | 117.3% | 21.2% |
| High Mobility | 381.0% | 311.4% | 149.6% | 23.4% |
| Uni-directional 1 | 626.2% | 12.4% | 475.3% | 99.3% |
| Uni-directional 2 | 349.2% | 337.4% | 32.2% | 20.8% |

These results suggest, as may be expected, that the largest reduction occurs when no intervention is in place. In the case, of implementing intervention in only one patch, the largest reduction occurs when the intervention is implemented in patch 1 only in all cases except for uni-directional 1. This might be so since $\beta_1 < \beta_2$. Finally, a strategy implementing maximum prevention for all time, may not be the most optimal strategy when using (6) as a 'metric'. For this strategy, the optimal control solution $(u^*, v^*)$ yields the largest reduction on the polar case and the smallest reduction on the asymmetric case.

The control functions shed light on the allocation of preventive resources (facemasks, disinfectants, sanitizers, and etc.) for reducing the number of infections, for example. Consider Figure 9. Under the conditions defined for uni-direction 1 case, most of the the resources should be allocated to Patch 1 as opposed to Patch 2. In both patches, the amount of resources is reduced as time increases to 60 day limit. The resources devoted to Patch 1 are allocated at a maximum for close to half of the 60 day period. The resources will not be devoted at a maximum at any point in the 60 day period to Patch 2, and they will decrease over the entire period. Under the conditions for uni-direction 2, most of the resources should be allocated to Patch 2, as opposed to Patch 1. Furthermore, the preventive resources will be allocated at a maximum in Patch 2 for almost the entire duration of the 60 day period. The resources for Patch 1 will increase at the onset of the epidemic outbreak, but gradually reduce as the 60 day period comes to an end. The resources in Patch 1 will never be devoted at a maximum over this period of time. We can make similar conclusions from the results presented in Figures 7 and

8. In all of these instances, these optimal strategies reduced the number of infections in both patches, as shown in Figures 7–9.

The results in Figures 7–9 further demonstrate strategic allocations of resources for reducing infected individuals. Depending on the scenario, the resources are allocated differently in each patch. Additionally, these strategies results show there is a reduction of the infected population as opposed to when no preventive resources are allocated. We compute the basic reproductive number using (4) and obtain $\mathcal{R}_0 > 1$ for all results in Figures 7–9. This suggests the optimal control functions, or equivalently the optimal allocation of resources, averted an epidemic outbreak in each patch.

While Figures 7–9 compared the optimal control solutions to scenarios without control, we also consider different scenarios for allocating preventive resources. Table 3 summarizes these results. The table further demonstrates that the controls obtained by solving (A1) are optimal in comparison to different strategies for the allocation of resources. The table suggests that the optimal control solution would outperform other strategies represented by different control scenarios, not just when preventive resources are not allocated.

## 4. Discussions

We have investigated the transmission dynamics in a two-patch SIR system. It is assumed that the two patches represent two locations that have a well-defined visiting relationship modeled by a residence-time matrix. The entries of the residence-time matrix depend on the infected individuals and model a behavioral response due to risk perception. We have developed an optimal control framework to identify optimal patch-specific control strategies under various virtual dispersal scenarios. First, we identify the optimal strategy to prevent or mitigate epidemics and compare with the results in the absence of controls. Our results indicate that, as expected, controlling the two patches simultaneously gives the best reduction in the total final epidemic sizes. Additionally, we found that the controlled two-patchy dynamics are strongly dependent on the following three key factors: virtual dispersal scenarios, transmission rates, the relative costs. Overall, controlling the outbreaks is more difficult as the transmission rates and relative cost increase.

**Author Contributions:** Conceptualization, S.L. and L.M.; methodology, O.B.; validation, S.L. and L.M.; formal analysis, O.B.; writing—original draft preparation, S.L.; writing—review and editing, L.M.; visualization, O.B.; supervision, S.L.; All authors have read and agreed to the published version of the manuscript.

**Funding:** S.L. was supported by the National Research Foundation of Korea (NRF) grant funded by the Korean government (MSIP) (NRF-2018R1A2B6007668).

**Conflicts of Interest:** The authors declare no conflict of interest.

## Appendix A

The optimal problem for the two-patch model is formulated to minimize the number of infected individuals in both patches for a finite time interval at a minimal cost of implementation. We define our objective functional as follows

$$J(u_1, u_2) = \int_0^{t_f} I_1 + I_2 + \frac{1}{2}(W_1 u_1^2 + W_2 u_2^2) \, dt$$

Then, we seek an optimal pair $(U^*, X^*)$ such that

$$J(U^*) = \min_{U \in \Omega} J(U), \tag{A1}$$

where $\Omega = \{(u_1, u_2) \in \left(L^1(0, t_f)\right)^2 \mid a \le u_i \le b, \ i = 1, 2\}$ subject to the state equations in (5) with $X = (S_1, I_1, R_1, S_2, I_2, R_2)$ and $U = (u_1, u_2)$. The existence of optimal controls is guaranteed by standard results of optimal control theory [20]. The necessary conditions of optimal solutions are

derived from Pontryagin's Maximum Principle [21]. This principle converts the system (5) into the problem of minimizing the Hamiltonian $H$ given by

$$
\begin{aligned}
H \;=\;& I_1 + I_2 + \frac{1}{2}(W_1 u_1^2 + W_2 u_2^2) + \lambda_1[-VS_1 I_1 - YS_1 I_2] + \lambda_2[VS_1 I_1 + YS_1 I_2 - \alpha_1 I_1] \quad \text{(A2)} \\
+\;& \lambda_3[-ZS_2 I_2 - YS_2 I_1] + \lambda_4[ZS_2 I_2 + YS_2 I_1 - \alpha_2 I_2]
\end{aligned}
$$

where

$$
\begin{aligned}
V = V(I_1, I_2) \;&=\; \frac{\beta_1(1 - u_1)p_{11}^2}{p_{11}N_1 + p_{21}N_2} + \frac{\beta_2(1 - u_2)p_{12}^2}{p_{12}N_1 + p_{22}N_2}, \\
Y = Y(I_1, I_2) \;&=\; \frac{\beta_1(1 - u_1)p_{11}p_{21}}{p_{11}N_1 + p_{21}N_2} + \frac{\beta_2(1 - u_2)p_{12}p_{22}}{p_{12}N_1 + p_{22}N_2}, \qquad \text{(A3)} \\
Z = Z(I_1, I_2) \;&=\; \frac{\beta_1(1 - u_1)p_{21}^2}{p_{11}N_1 + p_{21}N_2} + \frac{\beta_2(1 - u_2)p_{22}^2}{p_{12}N_1 + p_{22}N_2}
\end{aligned}
$$

are positive functions of $I_1$ and $I_2$. We present the following partial derivatives as they will be used in the proof of Theorem A1:

$$
\begin{aligned}
\frac{\partial V}{\partial I_1} &= A_1 + A_2, \quad \frac{\partial V}{\partial I_2} = A_3 + A_4, \\
\frac{\partial Y}{\partial I_1} &= B_1 + B_2, \quad \frac{\partial Y}{\partial I_2} = B_3 + B_4, \qquad \text{(A4)} \\
\frac{\partial Z}{\partial I_1} &= C_1 + C_2, \quad \frac{\partial Z}{\partial I_2} = C_3 + C_4,
\end{aligned}
$$

where

$$
\begin{aligned}
A_1 &= \frac{\beta_1(1 - u_1)\left[(p_{11}^2 N_1 + 2p_{11}p_{21}N_2)\frac{\partial p_{11}}{\partial I_1} - p_{11}^2 N_2 \frac{\partial p_{21}}{\partial I_1}\right]}{(p_{11}N_1 + p_{21}N_2)^2}, \\
A_2 &= \frac{\beta_2(1 - u_2)\left[(p_{12}^2 N_1 + 2p_{12}p_{22}N_2)\frac{\partial p_{12}}{\partial I_1} - p_{12}^2 N_2 \frac{\partial p_{22}}{\partial I_1}\right]}{(p_{12}N_1 + p_{22}N_2)^2}, \qquad \text{(A5)} \\
A_3 &= \frac{\beta_1(1 - u_1)\left[(p_{11}^2 N_1 + 2p_{11}p_{21}N_2)\frac{\partial p_{11}}{\partial I_2} - p_{11}^2 N_2 \frac{\partial p_{21}}{\partial I_2}\right]}{(p_{11}N_1 + p_{21}N_2)^2}, \\
A_4 &= \frac{\beta_2(1 - u_2)\left[(p_{12}^2 N_1 + 2p_{12}p_{22}N_2)\frac{\partial p_{12}}{\partial I_2} - p_{12}^2 N_2 \frac{\partial p_{22}}{\partial I_2}\right]}{(p_{12}N_1 + p_{22}N_2)^2},
\end{aligned}
$$

$$
\begin{aligned}
B_1 &= \frac{\beta_1(1 - u_1)\left[p_{21}^2 N_2 \frac{\partial p_{11}}{\partial I_1} + p_{11}^2 N_1 \frac{\partial p_{21}}{\partial I_1}\right]}{(p_{11}N_1 + p_{21}N_2)^2}, \\
B_2 &= \frac{\beta_2(1 - u_2)\left[p_{22}^2 N_2 \frac{\partial p_{12}}{\partial I_1} + p_{12}^2 N_1 \frac{\partial p_{22}}{\partial I_1}\right]}{(p_{12}N_1 + p_{22}N_2)^2}, \qquad \text{(A6)} \\
B_3 &= \frac{\beta_1(1 - u_1)\left[p_{21}^2 N_2 \frac{\partial p_{11}}{\partial I_2} + p_{11}^2 N_1 \frac{\partial p_{21}}{\partial I_2}\right]}{(p_{11}N_1 + p_{21}N_2)^2}, \\
B_4 &= \frac{\beta_2(1 - u_2)\left[p_{22}^2 N_2 \frac{\partial p_{12}}{\partial I_2} + p_{12}^2 N_1 \frac{\partial p_{22}}{\partial I_2}\right]}{(p_{12}N_1 + p_{22}N_2)^2},
\end{aligned}
$$

$$C_1 = \frac{\beta_1(1-u_1)\left[(2p_{11}p_{21}N_1 + p_{21}^2N_2)\frac{\partial p_{21}}{\partial I_1} - p_{21}^2N_1\frac{\partial p_{11}}{\partial I_1}\right]}{(p_{11}N_1 + p_{21}N_2)^2},$$

$$C_2 = \frac{\beta_2(1-u_2)\left[(2p_{22}p_{12}N_1 + p_{22}^2N_2)\frac{\partial p_{22}}{\partial I_1} - p_{12}^2N_1\frac{\partial p_{21}}{\partial I_1}\right]}{(p_{12}N_1 + p_{22}N_2)^2},$$

$$C_3 = \frac{\beta_1(1-u_1)\left[(2p_{11}p_{21}N_1 + p_{21}^2N_2)\frac{\partial p_{21}}{\partial I_2} - p_{21}^2N_1\frac{\partial p_{11}}{\partial I_2}\right]}{(p_{11}N_1 + p_{21}N_2)^2},$$

$$C_4 = \frac{\beta_2(1-u_2)\left[(2p_{22}p_{12}N_1 + p_{22}^2N_2)\frac{\partial p_{22}}{\partial I_2} - p_{22}^2N_1\frac{\partial p_{12}}{\partial I_2}\right]}{(p_{12}N_1 + p_{22}N_2)^2},$$

(A7)

with

$$\frac{\partial p_{11}}{\partial I_1} = \frac{(\sigma_{11}-1)I_2}{(1+I_1+I_2)^2}, \quad \frac{\partial p_{22}}{\partial I_1} = \frac{(1-\sigma_{22})(1+I_2)}{(1+I_1+I_2)^2}$$

$$\frac{\partial p_{12}}{\partial I_1} = \frac{\sigma_{12}I_2}{(1+I_1+I_2)^2}, \quad \frac{\partial p_{21}}{\partial I_1} = \frac{-\sigma_{21}(1+I_2)}{(1+I_1+I_2)^2},$$

$$\frac{\partial p_{11}}{\partial I_2} = \frac{(1-\sigma_{11})(1+I_1)}{(1+I_1+I_2)^2}, \quad \frac{\partial p_{22}}{\partial I_2} = \frac{(\sigma_{22}-1)I_1}{(1+I_1+I_2)^2},$$

$$\frac{\partial p_{12}}{\partial I_2} = \frac{-\sigma_{12}(1+I_1)}{(1+I_1+I_2)^2}, \quad \frac{\partial p_{21}}{\partial I_2} = \frac{\sigma_{21}I_1}{(1+I_1+I_2)^2}.$$

(A8)

Using Hamiltonian $H$ in (A2) and Pontryagin's Maximum Principle [21], we have the theorem:

**Theorem A1.** *There exist optimal controls $U^*$ and corresponding state solutions $X^*$ that minimize $J(U)$ over $\Omega$. In order for the above statement to be true, it is necessary that there exist continuous functions $\lambda_i = \lambda_i(t)$ such that*

$$\lambda_1' = -\frac{\partial H}{\partial S_1} = (\lambda_1 - \lambda_2)(VI_1 + YI_2),$$

$$\lambda_2' = -\frac{\partial H}{\partial I_1} = -1 + (\lambda_1 - \lambda_2)\left(\frac{\partial V}{\partial I_1}S_1I_1 + VS_1 + \frac{\partial Y}{\partial I_1}S_1I_2\right) + \alpha_1\lambda_2$$
$$+ (\lambda_3 - \lambda_4)\left(\frac{\partial Z}{\partial I_1}S_2I_2 + YS_2 + \frac{\partial Y}{\partial I_1}S_2I_1\right),$$

$$\lambda_3' = -\frac{\partial H}{\partial S_2} = (\lambda_3 - \lambda_4)(ZI_2 + YI_1),$$

$$\lambda_4' = -\frac{\partial H}{\partial I_2} = -1 + (\lambda_1 - \lambda_2)\left(\frac{\partial V}{\partial I_2}S_1I_1 + YS_1 + \frac{\partial Y}{\partial I_2}S_1I_2\right) + \alpha_2\lambda_4$$
$$+ (\lambda_3 - \lambda_4)\left(\frac{\partial Z}{\partial I_2}S_2I_2 + ZS_2 + \frac{\partial Y}{\partial I_2}S_2I_1\right),$$

(A9)

*with transversality conditions $\lambda_i(t_f) = 0, i = 1, \ldots 4$ and characterization of the optimal controls*

$$u_1 = \frac{1}{(p_{11}N_1 + p_{21}N_2)W_1}\left[-\lambda_1(\beta_1 p_{11}^2 S_1 I_1 + \beta_1 p_{11}p_{21}S_1 I_2) + \lambda_2(\beta_1 p_{11}^2 S_1 I_1 + \beta_1 p_{11}p_{21}S_1 I_2)\right.$$
$$- \lambda_3(\beta_1 p_{21}^2 S_2 I_2 + \beta_1 p_{11}p_{21}S_2 I_1) + \lambda_4(\beta_1 p_{21}^2 S_2 I_2 + \beta_1 p_{11}p_{21}S_2 I_1)\Big],$$

$$u_2 = \frac{1}{(p_{12}N_1 + p_{22}N_2)W_2}\left[\lambda_1(\beta_2 p_{12}^2 S_1 I_1 + \beta_2 p_{12}p_{22}S_1 I_2) + \lambda_2(\beta_2 p_{12}^2 S_1 I_1 + \beta_2 p_{12}p_{22}S_1 I_2)\right.$$
$$+ \lambda_3(\beta_2 p_{22}^2 S_2 I_2 + \beta_2 p_{12}p_{22}S_2 I_1) + \lambda_4(\beta_2 p_{22}^2 S_2 I_2 + \beta_2 p_{12}p_{22}S_2 I_1)\Big].$$

(A10)

**Proof.** The existence of optimal controls follows from Corollary 4.1 of [20] since the integrand of $J$ is a convex function in $U$ and the the state system satisfies the *Lipschitz* property with respect to the state variables. The following can be derived from the Pontryagin's Maximum Principle [21]:

$$\frac{d\lambda_1}{dt} = -\frac{\partial H}{\partial S_1}, \frac{d\lambda_2}{dt} = -\frac{\partial H}{\partial I_1}, \frac{d\lambda_3}{dt} = -\frac{\partial H}{\partial S_2}, \frac{d\lambda_4}{dt} = -\frac{\partial H}{\partial I_2},$$

with transversality condition $\lambda_i(t_f) = 0$ for $i = 1, ..., 4$, evaluated at the optimal controls and corresponding states. Differentiating $H$ with respect to $u_i$ and using (A4)–(A8), we obtain the equations

$$
\begin{aligned}
\frac{\partial H}{\partial u_1} &= W_1 u_1 + \lambda_1 S_1 \left( -\frac{\partial V}{\partial u_1} I_1 - \frac{\partial Y}{\partial u_1} I_2 \right) + \lambda_2 S_1 \left( \frac{\partial V}{\partial u_1} I_1 + \frac{\partial Y}{\partial u_1} I_2 \right) \\
&\quad + \lambda_3 S_2 \left( -\frac{\partial Z}{\partial u_1} I_2 - \frac{\partial Y}{\partial u_1} I_1 \right) + \lambda_4 S_2 \left( \frac{\partial Z}{\partial u_1} I_2 + \frac{\partial Y}{\partial u_1} I_1 \right) \\
&= 0, \\
\frac{\partial H}{\partial u_2} &= W_2 u_2 + \lambda_1 S_1 \left( -\frac{\partial V}{\partial u_2} I_1 - \frac{\partial Y}{\partial u_2} I_2 \right) + \lambda_2 S_1 \left( \frac{\partial V}{\partial u_2} I_1 + \frac{\partial Y}{\partial u_2} I_2 \right) \\
&\quad + \lambda_3 S_2 \left( -\frac{\partial Z}{\partial u_2} I_2 - \frac{\partial Y}{\partial u_2} I_1 \right) + \lambda_4 S_2 \left( \frac{\partial Z}{\partial u_2} I_2 + \frac{\partial Y}{\partial u_2} I_1 \right) \\
&= 0,
\end{aligned}
\tag{A11}
$$

where

$$
\begin{aligned}
\frac{\partial V}{\partial u_1} &= \frac{-\beta_1 p_{11}^2}{p_{11} N_1 + p_{21} N_2}, & \frac{\partial V}{\partial u_2} &= \frac{-\beta_2 p_{12}^2}{p_{12} N_1 + p_{22} N_2}, \\
\frac{\partial Y}{\partial u_1} &= \frac{-\beta_1 p_{11} p_{21}}{p_{11} N_1 + p_{21} N_2}, & \frac{\partial Y}{\partial u_2} &= \frac{-\beta_2 p_{12} p_{22}}{p_{12} N_1 + p_{22} N_2}, \\
\frac{\partial Z}{\partial u_1} &= \frac{-\beta_1 p_{21}^2}{p_{11} N_1 + p_{21} N_2}, & \frac{\partial Z}{\partial u_2} &= \frac{-\beta_2 p_{22}^2}{p_{12} N_1 + p_{22} N_2}.
\end{aligned}
$$

Solving for $u_i$ in (A11) yields the characterization for the control functions given in (A10). □

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
