# Peer review of "Resource Allocation in Two-Patch Epidemic Model with State-Dependent Dispersal Behaviors Using Optimal Control"

_processes, doi:10.3390/pr8091087_

Round 1

Reviewer 1 Report

The overall idea of this paper is interesting, but its clarity needs much improvement.

Using only references 15 and 16 for multi-patches is inadequate. See for example, a fundamental paper connecting Ro to multi-patches in Hsieh, van den Driessche, and Wang, 2007 Bulletin of Math Biology. The sentence about references 15 and 16 is not clear to say what is in each paper.

At the top of page 2, residents moving to reside in  the other patch is mentioned, but is that happening in their model?  Clarify the meaning of virtual dispersal.

The description of the model depends too much on reference 15.  Explain the sigmas.  The wording should be more careful  ...susceptible ...can be infected by a proportion....   should be ... susceptible can be infected an infected individual in proportion      With the last line on page 3, explain 'while in patch 1' and 'while in patch 2'.  Is I_1 the one changing patches there?  Tell some details about the overall system.  There are no births or deaths;  you say 'no vital dynamics' but maybe you mean demographic dynamics.   There are no deaths due to disease.

Explain why the costs of the controls do not depend on the sizes of the populations.

Either describe the scenarios on page 4 or move table 1 to later in the paper where the scenarios are used.

page 5  What is meant by 'as stated before' of the limit of I as time gets large?

Is it not clear why the results in figures 2 - 4 need to presented with the p_ij coefficients.

The J values should be given for each figure in Figures 7-9.  Write a few more sentences describing the results of these figures.

Remove the bullet points on page 12. They are true  due to properties of the optimal control.   Remove the equation with 100 x at bottom of the page. Table 3  is good, but you need to write a few concluding sentences about the scenarios.

What is the point of Table 4?  It is not discussed.  How is Ro calculated for those cases, meaning what value is used for the controls?

Figures 10 and 11 show obvious results when costs are lowered. Be careful with the wording,  ...resources will never be devoted at a maximum....This must just happen because of the cost size.  What if the cost coefficients were 1? 

Typo... page 3 the entries of residence time matrix p_ij (I_1, I_2)

page 14  This a residence..   page 2    strategies .... are studied numerically
